# Chitosan and Its Derivatives: Preparation and Antibacterial Properties

**DOI:** 10.3390/ma16186076

**Published:** 2023-09-05

**Authors:** Anton R. Egorov, Anatoly A. Kirichuk, Vasili V. Rubanik, Vasili V. Rubanik, Alexander G. Tskhovrebov, Andreii S. Kritchenkov

**Affiliations:** 1Department of Human Ecology and Biolementology, RUDN University, 6 Miklukho-Maklaya St., 117198 Moscow, Russia; sab.icex@mail.ru (A.R.E.); kirichuk-aa@rudn.ru (A.A.K.); alexander.tskhovrebov@gmail.com (A.G.T.); 2Institute of Technical Acoustics NAS of Belarus, Ludnikova Prosp. 13, 210009 Vitebsk, Belarus; v.v.rubanik@tut.by (V.V.R.); iakustika@mail.ru (V.V.R.J.)

**Keywords:** chitosan, films, nanoparticles, nonwoven materials, hydrogels, antibacterial activity

## Abstract

This comprehensive review illuminates the various methods of chitosan extraction, its antibacterial properties, and its multifarious applications in diverse sectors. We delve into chemical, physical, biological, hybrid, and green extraction techniques, each of which presents unique advantages and disadvantages. The choice of method is dictated by multiple variables, including the desired properties of chitosan, resource availability, cost, and environmental footprint. We explore the intricate relationship between chitosan’s antibacterial activity and its properties, such as cationic density, molecular weight, water solubility, and pH. Furthermore, we spotlight the burgeoning applications of chitosan-based materials like films, nanoparticles, nonwoven materials, and hydrogels across the food, biomedical, and agricultural sectors. The review concludes by highlighting the promising future of chitosan, underpinned by technological advancements and growing sustainability consciousness. However, the critical challenges of optimizing chitosan’s production for sustainability and efficiency remain to be tackled.

## 1. Introduction

Chitosan, a biocompatible, biodegradable and bioactive aminopolysaccharide derived from chitin, has generated significant interest in various fields, particularly due to its remarkable antibacterial properties. This natural biopolymer finds extensive applications in diverse areas such as food preservation, biomedicine, cosmetics, water treatment, and agriculture. The contribution of chitosan to the medical field is especially great, e.g., as antibacterial gels [1], biosensors [2], antitumor systems [3], antioxidants [4], antifungal agents [5], anti-neurodegenerative systems [6] and a lot more. The beginning of the current paper provides an exhaustive review of the methods of chitosan extraction, focusing on their advantages, drawbacks, and potential for future improvements. The review then delves into the antibacterial activity of chitosan, exploring the various factors influencing its antimicrobial effectiveness, and presents the key chitosan-based antibacterial materials currently in use, including films, nanoparticles, nonwoven materials, and hydrogels. It also investigates the role of chitosan’s inherent structural and physicochemical properties, the characteristics of targeted microorganisms, and environmental conditions in shaping its antimicrobial properties. A detailed examination of the interaction between chitosan’s cationic density, molecular weight, water solubility, and pH, and their consequent influence on its antibacterial potency is conducted. Through this review, the authors aim to provide a comprehensive understanding of the current state of chitosan production and its antibacterial applications, underscoring the need for continued research and development in the pursuit of more sustainable, efficient, and tailored methods for chitosan production. The uniqueness of this review is primarily in the balance between classics and innovation (between links to time-tested highly cited publications and citations of completely new but interesting works). In this regard, this review may even serve an educational function for students.

## 2. Sources of Chitosan

Chitosan is a biodegradable, biocompatible, and non-toxic natural aminopolysaccharide derived mainly from chitin, the second most abundant biopolymer on Earth after cellulose. Chitin can be found in various natural sources, and the type of source can influence the properties of the derived chitosan. Herein, we will highlight the primary sources of chitosan and briefly discuss their advantages and disadvantages.

Shellfish. The most common source of chitosan is the exoskeleton of shellfish such as crabs, shrimp, and lobsters.Advantages:Abundance: the seafood industry generates a substantial amount of shellfish waste, making this a readily available and inexpensive source of chitin;High yield: shellfish shells typically have high chitin content, leading to high yields of chitosan.Disadvantages:Allergenicity: shellfish is a common allergen, and there are concerns that chitosan derived from this source could potentially trigger allergic reactions;Environmental impact: the extraction process often uses harsh chemicals, which can have a negative environmental impact [7,8,9,10].Fungi. Certain types of fungi, particularly Aspergillus and Mucor species, are known to produce chitin in their cell walls.Advantages:No allergenicity: fungal-derived chitosan is less likely to provoke allergies compared with shellfish-derived chitosan;Sustainability: fungi can be cultivated in a controlled environment using waste materials, making this a sustainable source of chitosan.Disadvantages:Lower yield: fungal cell walls contain less chitin than shellfish shells, leading to a lower yield of chitosan;Complexity: the cultivation and extraction process can be complex and time-consuming [11,12,13,14,15].Insects. Insects like beetles, ants, and butterflies also contain chitin in their exoskeletons.Advantages:Variety: different insects produce chitin with different properties, which can be exploited to produce a wide range of chitosan products;Sustainability: insects have a high reproduction rate and require less space and resources compared with other animals.Disadvantages:Scale: it may be challenging to collect a large amount of insect biomass, making large-scale chitosan production difficult;Acceptability: there might be social and cultural barriers to using insects as a source of chitosan [16,17,18,19,20].Krill. Krill, small marine crustaceans, are also a viable source of chitosan.Advantages:Quality: krill-derived chitosan often has high solubility and low molecular weight, making it suitable for various applications.Disadvantages:Environmental impact: overharvesting of krill for chitosan production could have negative impacts on marine ecosystems.

While chitosan can be derived from a variety of sources (as depicted in Figure 1), each source presents its distinct set of strengths and weaknesses, encompassing yield, allergenic potential, environmental consequences, and sustainability aspects. When opting for a chitosan source for particular applications, it becomes imperative to carefully weigh these factors.

To pave the way for a greener and more responsible future, forthcoming research endeavors should be directed toward refining extraction techniques. Furthermore, the exploration of alternative chitosan sources holds promise, aiming not only to mitigate environmental repercussions but also to amplify sustainability efforts.

## 3. Methods of Obtaining Chitosan: Segments for Exploration

The processes for chitosan extraction from chitin can be categorized into five principal classifications: chemical, biological, physical, hybrid, and green extraction methods, as illustrated in Figure 2. Each method possesses distinct merits and limitations. Moreover, these methods can be employed individually or synergistically in combination to harness their collective benefits.

### 3.1. Chemical Methods

Conventional chemical methodologies for chitosan extraction from chitin entail the utilization of potent alkali solutions to expedite the deacetylation process. This operation prompts the elimination of acetyl groups from chitin, culminating in the creation of chitosan.

Acidic Deacetylation. This technique encompasses treating chitin with acids like hydrochloric acid (HCl) or sulfuric acid (H_2_SO_4_). The acids aid in disassembling the crystalline arrangement of chitin, heightening its responsiveness. Nonetheless, this method’s efficacy in achieving comprehensive deacetylation is limited, potentially resulting in the formation of low-molecular-weight chitosan owing to the aminopolysaccharide backbone’s vulnerability to acid hydrolysis. Thus, treatment with strong acids is used for crustacean shells, but this is used for dissolving the carbonate component of the exoskeleton rather than for the hydrolysis of amide bonds [21,22].Alkaline Deacetylation. This method is the most commonly used for chitosan production. It involves treating chitin with strong alkalis, typically sodium hydroxide (NaOH) or potassium hydroxide (KOH), at high temperatures for extended periods. The alkali breaks the acetyl–amino linkage in chitin, causing the release of acetate ions and the formation of chitosan [23]. The extent of deacetylation, which can be controlled by the concentration of alkali, temperature, and duration of the reaction, determines the properties of the resulting chitosan [24]. While this method is highly effective and can yield chitosan with a high degree of deacetylation, it often requires large quantities of alkali and high energy input due to the need for elevated temperatures, thus raising environmental concerns. Additionally, harsh alkaline conditions can cause depolymerization, leading to chitosan with lower molecular weight [25].

The conditions under which the chemical deacetylation reaction takes place play a significant role in determining the efficiency of the process and the properties of the resulting chitosan.

*Temperature:* The rate of deacetylation increases with temperature, resulting in a higher degree of deacetylation. However, excessive heat can lead to the degradation of chitosan [25];*Time:* Longer reaction times can lead to a higher degree of deacetylation. However, prolonged exposure to harsh conditions can also cause depolymerization, resulting in low-molecular-weight chitosan [25];*Concentration:* Using higher concentrations of alkali can result in a higher degree of deacetylation. However, higher concentrations can also cause more significant depolymerization, leading to lower-molecular-weight chitosan [25].

In essence, although chemical extraction methods for chitosan have proved effective, their application necessitates the meticulous management of reaction parameters and frequently demands substantial quantities of chemicals and energy. This circumstance has spurred notable enthusiasm for the advancement of extraction approaches that align with the principles of sustainability and enhanced efficiency.

### 3.2. Biological Methods

Biological techniques for acquiring chitosan from chitin encompass the employment of enzymes and microorganisms adept at naturally deacetylating chitin. In contrast to chemical methodologies, these approaches are generally more ecologically sound. They necessitate reduced energy consumption and result in fewer perilous waste byproducts, rendering them environmentally favorable.

Enzymatic Deacetylation. Enzymatic deacetylation involves the use of chitinase and other enzymes to convert chitin to chitosan. Chitinase, produced by a variety of bacteria, fungi, and plants, breaks down chitin by cleaving the glycosidic bonds in the polymer. Chitosanase, on the other hand, can further deacetylate chitosan, adjusting its degree of deacetylation [26]. While enzymatic deacetylation is more eco-friendly and offers more control over the properties of the resulting chitosan, it is often slower and more expensive than chemical methods due to the cost of enzymes and the need for precise control over reaction conditions [27]. Furthermore, not all enzymes can access and cleave the bonds in highly crystalline regions of chitin, limiting the degree of deacetylation that can be achieved [28,29,30].Microbial Deacetylation. Microbial deacetylation involves the use of microorganisms, such as bacteria and fungi, to deacetylate chitin. Several strains of bacteria, including *S. marcescens* and *P. aeruginosa*, as well as fungi like *A. niger* and *M. rouxii*, have been found to produce enzymes that can deacetylate chitin [31]. The advantage of microbial deacetylation is that it can be carried out at near-ambient temperatures and pressures, reducing energy requirements compared with chemical methods. However, it is typically slower and may require specific growth media and conditions for the microorganisms, which can increase the complexity and cost of the process. Microbial and enzymatic deacetylation also offer the possibility of bio-selective deacetylation, which could lead to chitosan with specific patterns of acetylation. This could open up new applications for chitosan that are not possible with randomly deacetylated chitosan produced through chemical methods [10,32].

Consequently, biological means of chitosan procurement offer a more sustainable substitute to chemical techniques. However, certain challenges warrant attention, encompassing the expense, pace, and level of deacetylation attainable through these avenues. Progress in biotechnology, coupled with an enhanced comprehension of the enzymes and microorganisms pivotal in the deacetylation process, holds the potential to surmount these obstacles in the forthcoming years.

### 3.3. Physical Methods

Physical approaches for chitosan extraction from chitin encompass manipulating chitin’s physical attributes to facilitate the deacetylation procedure. These techniques can be employed individually or in conjunction with chemical or biological methods to enhance overall efficiency. Here are some of the primary physical methods commonly employed in chitosan extraction:Mechanical Treatment. Mechanical methods such as grinding, milling, and sonication are often used to physically disrupt the chitin structure and increase its reactivity. This enhances its accessibility to chemical or enzymatic treatments, thereby facilitating the deacetylation process [33].
Grinding/Milling. These methods reduce the particle size of chitin, increasing its surface area and making it more accessible for deacetylation. However, the process can generate heat and induce changes in the chitin structure that may affect the properties of the resulting chitosan [34];Sonication. This involves the use of ultrasonic waves to disrupt the chitin structure. Sonication can enhance the penetration of deacetylating agents into the chitin structure, improving the efficiency of the deacetylation process. However, the process requires specialized equipment and may also induce changes in the chitin structure [35,36].
Thermal Treatment. Thermal methods involve the use of heat to facilitate the deacetylation process. High temperatures can enhance the reactivity of chitin and speed up the deacetylation process. However, excessive heat can lead to degradation of the chitosan product [37].Irradiation. Radiation methods involve the use of microwave, UV, or gamma radiation to facilitate the deacetylation process.
Microwave-assisted deacetylation. This approach uses microwave radiation to heat chitin, accelerating the deacetylation process. Microwave radiation can improve the efficiency and reduce the duration of the deacetylation process [38];UV irradiation. These techniques use radiation to break down the chitin structure, making it more accessible for deacetylation. However, the high energy associated with this process may lead to changes in the chitin structure and properties of the resulting chitosan [39].


Physical methods can improve the efficiency of the deacetylation process, but they often require additional energy input and can induce changes in the chitin structure that affect the properties of the resulting chitosan. They are generally used in combination with chemical or biological methods to enhance their effectiveness. The authors note that it is disputable to put physical methods at the same level as chemical ones, rather than as auxiliary ones. Chitin-to-chitosan conversion requires a (bio)chemical reaction, and physical processing can only assist it.

### 3.4. Hybrid Methods

Hybrid strategies for chitosan extraction from chitin encompass amalgamating chemical, physical, and biological techniques to optimize the deacetylation process’s efficiency. These integrated methodologies frequently surmount certain constraints associated with individual methods, culminating in enhanced yield and superior quality of the resulting chitosan.

Chemical–Physical Hybrid Methods. This combination generally involves a mechanical or thermal pre-treatment of chitin to increase its reactivity, followed by a chemical deacetylation step. The pre-treatment disrupts the crystalline structure of chitin, making it more susceptible to the chemical deacetylation process. This can lead to a higher degree of deacetylation and a higher yield of chitosan [22];Chemical–Biological Hybrid Methods. These methods combine the strengths of chemical and biological processes. Typically, a mild chemical treatment is used initially to increase the reactivity of chitin, and then a biological process (either enzymatic or microbial) is used for deacetylation. This approach can enhance the efficiency of the deacetylation process, reduce the environmental impact associated with chemical methods, and provide more control over the properties of the resulting chitosan [40];Physical–Biological Hybrid Methods. In these cases, a physical pre-treatment step (such as grinding, sonication, or irradiation) is used to disrupt the chitin structure and increase its accessibility to enzymes or microorganisms. This can improve the efficiency of the biological deacetylation process, reducing the time and cost required to produce chitosan [41];Tri-hybrid Methods. In some cases, all three methods (chemical, physical, and biological) may be combined to maximize the efficiency of the deacetylation process. For example, a physical pre-treatment could be used to disrupt the chitin structure, followed by a mild chemical treatment to increase reactivity, and, finally, a biological process to deacetylate chitin [42].

Although hybrid techniques exhibit the potential to elevate the efficiency, yield, and oversight in chitosan production, they often mandate intricate process management and could entail elevated costs due to the incorporation of multiple treatments. Nonetheless, through meticulous refinement, these approaches hold promise in generating premium-grade chitosan with heightened sustainability and efficiency.

### 3.5. Green Extraction Methods

Green extraction methods for chitosan endeavor to mitigate the environmental repercussions linked to conventional techniques, which frequently entail the deployment of aggressive chemicals and substantial energy consumption. These environmentally conscious approaches can be classified as follows:Supercritical or Subcritical Fluid-Based Processes. These methods of chitosan extraction aim to eliminate or reduce the use of harsh chemicals and solvents. One such method involves using supercritical or subcritical fluids, such as supercritical carbon dioxide or water, to deacetylate chitin. These fluids can act as both a solvent and a deacetylating agent, reducing the need for additional chemicals. Supercritical fluid extraction is generally more energy-efficient than traditional methods and can result in chitosan with high purity [43];Use of Green Solvents. Green solvents are those that are less harmful to the environment, either because they are biodegradable, have low toxicity, or are derived from renewable resources. In the context of chitosan extraction, ionic liquids (salts in a liquid state) are often used as green solvents. They can effectively dissolve and deacetylate chitin under mild conditions, reducing the energy requirements of the process. However, the recovery and reuse of ionic liquids can be challenging and may offset their environmental benefits [44];Energy-efficient Methods. Energy-efficient methods aim to reduce the energy required to extract chitosan from chitin. This often involves optimizing the deacetylation process to operate at lower temperatures or shorter times or developing more efficient systems for heat and mass transfer. Microwave-assisted extraction, for example, can accelerate the deacetylation process and reduce energy consumption [45];Bio-based Methods. Bio-based methods use enzymes or microorganisms to deacetylate chitin, which typically require less energy and produce fewer hazardous waste products than chemical methods. Some bio-based methods even aim to integrate chitosan production into a biorefinery concept, where multiple valuable products are produced from biomass in a sustainable manner [46].

Even though green extraction methods for chitosan present substantial environmental advantages, certain challenges demand attention. These encompass enhancing the efficiency and scalability of these techniques, curbing associated expenses, and ensuring the caliber and uniformity of the resultant chitosan. Nonetheless, given the escalating apprehensions about environmental sustainability and the continual advancements in green chemistry and biotechnology, these methods are anticipated to assume a more prominent role in the landscape of chitosan production in the forthcoming era.

### 3.6. Comparison of Methods

The diverse techniques for chitosan extraction exhibit distinctive merits and drawbacks. The selection of the optimal method hinges on several factors, encompassing the sought-after attributes of chitosan, the accessible resources and equipment, cost implications, and environmental consequences. In this context, we present a concise comparative overview of the various methods:Chemical Methods. Chemical methods, particularly alkaline deacetylation, are the most commonly used methods for chitosan extraction due to their high efficiency and ability to produce chitosan with a high degree of deacetylation. However, they require large amounts of chemicals and energy, generate significant amounts of waste, and may lead to degradation of the chitosan. They also lack selectivity, resulting in chitosan with a random distribution of acetyl groups [47];Physical Methods. Physical methods can enhance the efficiency of the deacetylation process by increasing the reactivity of chitin. However, they often require additional energy input and can induce changes in the chitin structure that affect the properties of the resulting chitosan. They are generally used in combination with chemical or biological methods rather than as standalone methods [48];Biological Methods. Biological methods offer a more sustainable alternative to chemical methods, requiring less energy and producing fewer hazardous waste products. They also offer the possibility of selective deacetylation, which could lead to chitosan with specific patterns of acetylation. However, these methods are typically slower and may be more expensive due to the cost of enzymes or the need for specific growth media for microorganisms [49];Hybrid Methods. Hybrid methods combine the strengths of chemical, physical, and biological methods, providing improved efficiency and control over chitosan production. However, they often require more complex process control and may have higher costs associated with the use of multiple treatments [41];Green Extraction Methods. Green extraction methods aim to reduce the environmental impact of chitosan extraction by minimizing the use of harsh chemicals and energy. These methods can be more environmentally friendly and can produce chitosan with high purity. However, they may require specialized equipment or materials (such as green solvents or supercritical fluids), and their efficiency and scalability may need to be improved [44].

While the quest for chitosan extraction does not yet possess a universal solution, the escalating emphasis on sustainability and technological progress is propelling the drive toward more ecologically sound and efficacious methods. The ideal approach would not only yield chitosan attuned to specific requirements but also exhibit economic viability and a negligible ecological footprint.

The trajectory of chitosan production is likely to be shaped by a confluence of technological advancements, conscientious sustainability contemplations, and the exigency for fine-tuned, top-quality chitosan adaptable to diverse applications. Below, we delineate, based on our own perspective and observations, some plausible future directions in this realm:Advanced Biotechnology: The utilization of genetically modified organisms or engineered enzymes has the potential to significantly enhance the efficiency and precision of the biological deacetylation process. This advancement could pave the way for producing chitosan with meticulously controlled properties, thereby unlocking novel applications in fields like drug delivery, tissue engineering, and biotechnology;Green Chemistry: The ongoing evolution of green extraction techniques, encompassing the use of environmentally friendly solvents, supercritical fluids, and energy-efficient processes, is anticipated. Beyond merely reducing the environmental footprint of chitosan production, these approaches might also elevate the caliber and purity of the resultant chitosan;Process Optimization and Scale-up: It is imperative to delve further into the optimization and scaling of chitosan extraction processes, particularly within the realm of biological and green extraction methods. This undertaking entails bolstering the efficiency of the deacetylation process, curbing chitosan production costs, and devising scalable processes capable of accommodating the surging demand;Integrated Biorefinery Approaches: A burgeoning interest is being witnessed in integrating chitosan production into the framework of a biorefinery concept. This visionary strategy entails generating a range of valuable products from biomass in an environmentally sustainable manner. This could encompass the simultaneous production of chitosan and other high-value commodities from chitin-containing waste streams, thereby diminishing waste and enhancing the economic viability of chitosan production;Advanced Characterization Techniques: The pursuit of developing and applying sophisticated characterization methodologies has the potential to delve deeper into the intricacies of chitosan’s structure and properties. This deeper understanding facilitates the correlation between extraction methodologies and chitosan properties, thus guiding the discernment and optimization of extraction techniques.

We hold a firm conviction that the trajectory of chitosan production is poised for a promising ascent, replete with manifold prospects for innovation and enhancement. With the sustained momentum of research and development, it is foreseeable that more sustainable, efficient, and custom-tailored methods for chitosan production will materialize, thereby broadening the horizon of potential applications for this versatile biopolymer.

## 4. Antibacterial Activity of Chitosan and Its Derivatives

### 4.1. Mechanism of Antibacterial Effect

The antibacterial activity of chitosan and its derivatives is contingent upon a myriad of factors that can be categorized into three main groups:The inherent structural and physicochemical traits of chitosan, which include its molecular weight and distribution, cationic density, degree of deacetylation, and balance between hydrophilic and hydrophobic properties;The specific type and strain of the microorganism targeted by the chitosan;Various environmental conditions, such as ionic strength, pH, temperature, etc.

The scientific literature presents three primary models elucidating the antimicrobial mechanisms of chitosan. The first model places emphasis on the interaction between positively-charged macromolecules and the negatively-charged cell surface. The second model centers on the penetration of chitosan molecules into the interior of the cell. The third model highlights the chelation by chitosan of pivotal metal ions essential for bacterial survival. In the ensuing discussion, we will briefly explore these models:

#### 4.1.1. Model Focusing on the Interaction of a Polycation with Anionic Sites on the Bacterial Cell Surface

The first model primarily focuses on the interaction between the positively-charged chitosan polycation and negatively-charged segments of the microbial cell surface. This interaction is mediated by Coulombic electrostatic forces between the protonated amine groups of chitosan (NH_3_^+^) and negatively-charged residues on the surface of the bacterial cell. Moreover, the protonated amine groups of chitosan compete effectively even with Ca^2+^ ions, which, in conventional conditions, electrostatically interact with the negatively-charged areas of the bacteria surface. This electrostatic interaction of chitosan with bacterial surface and the displacement of Ca^2+^ ions lead to at least two adverse effects on the microbial cell:A significant alteration in membrane permeability properties, which induces an internal osmotic imbalance which can, finally, result in breakage of the bacterial cell wall;the hydrolytic breakdown of peptidoglycans in the microorganism’s cell wall, resulting in the leakage of intracellular content into the environment.

This model was meticulously examined in various studies. The adhesion of chitosan to the bacterial cell surface can be directly visualized using scanning electron microscopy or transmission electron microscopy [50]. Amorim et al. [51] performed ultrastructural analyses of the clinical isolates *S. aureus* and *E. coli* by transmission electron microscopy before and after chitosan treatment. They observed that chitosan effectively adheres to the surface of bacterial cells and causes the bacterial cell wall to rupture. Zhenzhen Zhang et al. used scanning electron microscopy and revealed that a novel cationic chitosan derivative, 3,6-*O*-[*N*-(2-aminoethyl)-acetamide-yl]-chitosan, kills bacteria by disrupting their membranes [52]. Similar results were obtained by Jenny Kim’s group, who confirmed that ultrastructural alterations in *P. acnes* cells were identified under the influence of positively-charged chitosan-based nanoparticles. They detected chitosan molecules attached to bacterial cell surfaces [53].

In addition to the direct method of microscopic observation of bacterial membrane rupture, a very convenient spectrophotometric method for observing this process is described in the literature. This approach is based on the following fact. If the antibacterial system can damage the membrane of the bacterial cell, then the contents of the bacterial cell leak into the external environment. This, in turn, results in a strong increase in adsorption at 260 nm. Using this approach, it was confirmed that chitosan and many of its cationic derivatives (triazole betaine [54], betainic- and sulfur-containing betaine derivatives [55,56], trimethylaminobenzyl [57], and many others) effectively destroy the integrity of the bacterial cell wall.

#### 4.1.2. Model Focusing on the Penetration of Chitosan Polycation into the Bacterial Cell

Another proposed mechanism involves chitosan binding to microbial DNA, which inhibits the protein synthesis cascade. It is clear that such a model requires the penetration of chitosan macromolecules into the cells of microorganisms [58,59]. It should be noted that only oligomeric molecules of chitosan can traverse the bacterial cell wall and reach the intracellular space. Studies using confocal laser scanning microscopy [60,61,62,63,64,65] have confirmed the presence of chitosan oligomers (chains with a small number of monomeric links) within *E. coli* cells. However, despite this model being accepted as a possible mechanism, the likelihood of its realization is considered significantly lower compared with the first model, even for chitosan oligomers [66]. The prevailing statement is that chitosan acts as a membrane disruptor rather than a penetrating substance [66,67].

#### 4.1.3. Model Focusing on Chelating by Chitosan Metal Ions

The third model posits that chitosan acts as an agent that chelates metal ions and binds cations necessary for microbial growth [68,69]. Chitosan exhibits a high chelating capacity for various metal ions (including Ni^2+^, Zn^2+^, Co^2+^, Fe^3+^, Mg^2+^ and Cu^2+^) and is widely used for the removal or extraction of metal ions in various industries [70]. Metal ions that are linked to the molecules of a microorganism’s cell wall through coordination and ionic interactions are crucial for the stability of the cell wall. Chitosan-mediated chelation of such metal ions is often associated with a possible form of antimicrobial action [71]. It is worth noting that the coordination binding of metal cations via chelation is facilitated by free non-protonated amino groups in chitosan molecules [72].

The structure of complexes of chitosan with metal cations can be different, and this very strongly depends on the degree of deacetylation of chitosan, its molecular weight and pH, the nature and oxidation state of the metal center, as well as on the molar ratio of chitosan and the metal cation [73,74,75,76,77,78]. In Figure 1, we show one of the possible structures of a coordination compound that includes chitosan as a macromolecular ligand [79].

Nonetheless, the antimicrobial mechanism of chitosan proposed in the third model appears to be of secondary significance. This is due to the fact that the amino groups available for coordinating interactions with metal centers are somewhat constrained at pH levels relevant to the infection process, and the formation of complexes attains saturation based on the concentration of metals.

Drawing from our extensive experience and contemplation spanning numerous years, we hold the view that the foremost mechanism underpinning the antibacterial efficacy of chitosan hinges on the electrostatic interaction between the cationic chitosan macromolecule and the negatively-charged sites on the bacterial cell membrane. The second and third models, while insightful and pertinent, should be regarded as supplementary and intriguing yet not supplant the primary significance of the first model.

### 4.2. The Effect of Cationic Density and Ways to Increase It

High cationic density in the macromolecules of chitosan and its derivatives leads to a strong electrostatic interaction with the negatively-charged segments of the bacterial surface [80,81]. The cationic density of chitosan macromolecules is determined by its degree of deacetylation [82]. The highest cationic density is characteristic for chitosans with the highest degree of deacetylation because it is the deacetylated free amino groups of chitosan that can undergo effective protonation, forming positively-charged NH_3_^+^ fragments [83]. For this reason, chitosans with a high degree of deacetylation have significantly greater antibacterial activity against both Gram-positive and Gram-negative bacteria compared with chitosans with a low degree of deacetylation [84].

Increasing the cationic density of chitosan can be achieved using two strategies: (1) increasing the degree of deacetylation; (2) introducing into the chitosan chain substituents containing cationic fragments. Using the second strategy results in derivatives of chitosan that typically possess greater antibacterial effectiveness than the original chitosan [85,86].

The main advantage of chitosan over other polysaccharides (cellulose, starch, carrageenan, alginic and hyaluronic acids, etc.) lies in the much greater ease of its chemical modification [87]. The presence of an amino group along with the primary alcohol function allows *N*-substituted, *O*-modified or *N*,*O*-substituted derivatives of chitosan to be obtained (Figure 2). Chitosan derivatization is primarily carried out to improve its physicochemical and biological properties. For example, *N*-quaternized chitosan has excellent water solubility and significantly greater antibacterial activity compared with starting chitosan, *N*-substituted tetrazole [88], triazole [89], azide [90], selenodiazole [86], oxadiazole [91], and many other derivatives. In fact, the literature describes significantly more examples of *N*-substituted antibacterial chitosan derivatives than *O*-substituted ones. This is clearly due to the greater reactivity of the amino group compared with the hydroxyl group (for example, the nucleophilic properties of the amino group are much greater than the nucleophilic ability of the OH group).

Figure 3 showcases instances of substituents introduced into the chitosan backbone through N-substitution, aimed at crafting exceptionally potent antibacterial derivatives. These illustrations provide a broad overview of the extensive array of chemical structures exhibited by these substituents.

Indeed, if chitosan is treated with an alkylating reagent, for example, diethylaminoethyl chloride, then under conventional conditions, the substitution occurs at both nucleophilic centers, with a significant predominance of *N*-substitution [110,111,112]. There are certain strategies for controlling this synthetic process that allow for the preparation of chitosan derivatives selectively at the desired reaction center with the desired degree of substitution. To selectively obtain *N*-substituted antibacterial chitosan derivatives, the pH of the reaction medium should be lowered to strongly acidic values. This approach leads to a sharp loss of the nucleophilic power of the hydroxyl group and only a slight decrease in the nucleophilicity of the amino group, that is, it has a differentiating effect [112]. If researchers aim to obtain selectively *O*-substituted polymers, then a synthetic strategy based on the protection of the amino group should be used [113,114,115,116]. The benzylidene protection of the amino group stands out for its high level of effectiveness and practical convenience in preparation. This remarkably straightforward synthetic process entails treating chitosan with an excess of benzaldehyde. Rapidly forming a Schiff base, this intermediate undergoes a subsequent reaction with an alkylating reagent via the hydroxyl group, yielding the intended O-derivative. Subsequently, a straightforward and swift deprotection step is executed through the familiar acid hydrolysis of the Schiff base. These well-established methodologies within chitosan chemistry are succinctly illustrated in Figure 3.

### 4.3. The Influence of Molecular Weight on the Antibacterial Effect

It can be argued that there is a relationship known as “the molecular weight of chitosan—the antibacterial effect”, even though, it seemed, not so long ago, that such a statement was only a crude hypothesis. Indeed, these correlations exist, but to this day, they cannot be called unambiguous. There are examples in the literature illustrating the increased activity of low-molecular-weight chitosan against *E. coli* compared with high-molecular-weight chitosan. However, there are absolutely opposite examples regarding the same bacterium [117]. For example, the literature reported no effect of the molecular weight of chitosan on the antibacterial activity against the same *E. coli* [118]. These seemingly contradictory results obviously suggest that the question of the influence of molecular weight on activity is a complex issue and requires an individual solution in each specific case.

The molecular weight of chitosan is an extremely important factor that makes it possible for this polymer to penetrate into a bacterial cell [119]. This is important for the development of the antibacterial effect, according to the second model. The cell wall of bacteria is a very reliable barrier that prevents the penetration of undesired external substances into the bacterial cell. In general, when a bacterial cell appears, either cases of disease (ionic pumps, clathrin-mediated endocytosis, etc.) or ordinary diffusion through the pores occur [83,120,121]. Proteins with a molecular weight of 50–100 kDa readily penetrate through the pores of a bacterial cell [122]. Chitosans of low molecular weight (50–100 kDa, hydrodynamic diameter about 50 nm) have also been detected through bacterial pores [123].

In summary, the role of molecular weight in chitosan’s antibacterial properties is significant, with various factors contributing to its effectiveness:Surface activity: smaller chitosan particles with a higher surface area-to-volume ratio exhibit enhanced surface activity, facilitating interactions with bacterial cells;Penetration: smaller particles can more readily breach bacterial cell walls, disrupting their functionality and inducing cell demise;Diffusion and dispersibility: smaller chitosan particles disperse and diffuse more effectively in solutions, leading to uniform distribution and improved contact with bacterial cells, thereby enhancing antibacterial action.

The discussed examples and rationale underline that, while the overarching trend suggests that smaller chitosan particles (lower molecular mass) possess heightened antibacterial effects, specific outcomes can hinge on diverse factors, encompassing bacteria type, pH levels, temperature, and experimental conditions. Consequently, refining the molecular weight of chitosan for optimal antibacterial applications often necessitates meticulous study and experimentation.

### 4.4. Evaluating the Interplay of Water Attraction and Repulsion and Solubility in Antibacterial Potency

Within the realm of antibacterial agents, especially those boasting a polymeric architecture, water takes on a pivotal role in orchestrating their antimicrobial functions. The foundation of this is the incapability of completely desiccated forms to unleash the stored energy intrinsic to their chemical structures, energy that is required for interaction with bacteria. This context converges seamlessly with the fact that bacteria, whether in in vitro or in vivo settings, invariably reside in contact with water. This dynamic sets the stage for the equilibrium between hydrophilic (water-attracting) and hydrophobic (water-repelling) traits of an antimicrobial agent to emerge as a fundamental driver of its functional mechanism and efficacy.

Chitosan, poised as a prospective contender in the realm of antibacterial weaponry, finds its solubility in water under strict governance by its inherently hydrophilic essence. Yet, lamentably, the inherent water solubility of chitosan trails on the side of inadequacy, erecting a barrier to its widespread adoption in the ambit of antibacterial applications [124].

Enhancing chitosan’s solubility through chemical alteration is a promising approach, offering the potential to widen its use as an antibacterial compound [125]. Thus, the pursuit of water-compatible chitosan and its derivative forms has become a crucial goal in antimicrobial research. Various chemical modification techniques like embedding mono- or oligosaccharide units into the chitosan structure, alkylation, acylation, quaternization, and metallization have been used to reach this objective. For example, one can obtain a quaternized version of chitosan, specifically its ammonium salt, by incorporating a quaternary ammonium group into the chitosan backbone. One notable study demonstrated that a quaternized chitosan variant, *N*,*N*,*N*-trimethylammonium chitosan chloride, had superior antibacterial capabilities, a wider spectrum of antimicrobial activity, and an accelerated rate of bacterial cell destruction than its unmodified counterpart [58,126,127].

Nevertheless, this is not to suggest that quaternized chitosan always outperforms in terms of antibacterial potency. It has been documented that integrating an *N*,*N*-dimethylaminobenzyl fragment or an *N*-pyridylmethyl group into the chitosan backbone did not boost antibacterial activity against *S. aureus* compared with the original chitosan [128]. Interestingly, even when these polymers showed a higher degree of quaternization than *N*,*N*,*N*-trimethylammonium chitosan chloride, their antibacterial effectiveness against *S. aureus* remained stagnant. This evidence illustrates that an increased level of *N*-substitution with *N*,*N*-dimethylaminobenzyl substituents can disrupt the hydrophilic–hydrophobic balance and lessen the likelihood of interaction between these chitosan variants and the bacterial cell wall. Similarly, Liu et al. elaborated highly antibacterial systems based on *p*-coumaric-acid-modified quaternized chitosan [129].

The antimicrobial potential of chitosan derivatives is significantly influenced by the size and characteristics of the spacer, given the consequential modifications to the polymer’s shape and charge density. Such modifications also alter its interaction mechanism with the cytoplasmic membrane [130]. In this context, the hydrophobic characteristics inherent to N-substituted chitosan confer a beneficial edge to the interaction between the polymer molecule and bacterial entities. An illustrative instance lies in N-hexadecanyl chitosan, the antimicrobial prowess of which was assessed against *E. coli*, *P. aeruginosa*, *S. aureus*, and *B. cereus*. Across the spectrum of scrutinized microorganisms, the antibacterial efficacy of these innovative alkyl derivatives surged notably beyond that of the initial chitosan. This augmentation finds its rationale in the hydrophobic nature of the introduced substituent, which engenders heightened antibacterial attributes [131].

Another study directed its focus on the potentiation of the antimicrobial influence of the low-molecular-weight derivative, namely mercaptoundecanoic-acid-grafted chitosan, alongside their corresponding nanoparticles loaded with carvacrol. This augmentation of antibacterial efficacy was ascribed to the hydrophobic aryl substituent’s role in expediting the derivative’s assimilation into bacterial cells. This, in turn, accentuated the hydrophobic interaction between the derivative and the cell, yielding an intensified antibacterial effect [132].

In a recent investigation, the hydrophobic attributes of chitosan underwent enhancement while its antibacterial efficacy was upheld through the grafting of dodecenylsuccinyl chains onto phthaloylchitosan, primarily targeting the C-6 position of the glucopyranose cyclic chain. The synthesis of dodecenylsuccinylated phthaloylchitosan was accomplished through a sequential process involving phthaloylation, dodecenylsuccinylation, and subsequent hydrazinolysis. This chemical alteration of chitosan yielded a remarkable surge in antibacterial potency specifically against Gram-positive bacteria.

A film derived from a solution of this chitosan derivative exhibited superior inhibition of bacterial growth and a more effective vapor barrier compared with a film composed of the unmodified chitosan. As a result, the authors of this study propose the application of this refined film as a coating for perishable food products, thereby augmenting their shelf life by inhibiting bacterial proliferation [133].

These observations find further affirmation in the following insight: the impact of acyl group length on antibacterial effectiveness becomes evident when hindering the growth of *E. coli* with acyl chitosan derivatives. The elevation in antibacterial activity attributed to acyl groups is likely grounded in the heightened hydrophobicity conferred by the acyl-substituted polymer. For instance, N-hexanoylchitosan sulfate, characterized by a more pronounced hydrophobic character due to its longer acyl chain (six carbon atoms), exhibited a more robust inhibitory effect in comparison to the relatively less hydrophobic N-propanoylchitosan sulfate (three carbon atoms) [134].

Chitosan’s solubility in water significantly impacts its antibacterial activity:Amino Groups and Solubility. Chitosan’s amino groups can be protonated, enabling them to dissolve in water, especially under acidic conditions;Molecular Weight Influence. Lower-molecular-weight chitosan tends to be more soluble in water, which can lead to better dispersion and increased interaction with bacterial cells;Antibacterial Activity. Higher solubility increases chitosan’s bioavailability and therefore its antibacterial activity.

However, why are water-soluble forms of chitosan characterized by a significantly greater antibacterial effect? The following explanation seems logical and reasonable to us. In their dissolved state, soluble chitosan and its cationic counterparts adopt a disassociated, lengthened structure. This particular shape allows for more effective engagement with the bacterial cell surface [135]. This underpins the superior antibacterial capabilities of soluble chitosan derivatives compared with those in insoluble forms.

Indeed, water solubility stands as a pivotal property of chitosan intricately linked to its antibacterial efficacy. This association underscores why chitosan derivatives or modifications are frequently engineered to enhance this characteristic, catering to distinct applications. Nevertheless, the interplay between chitosan’s solubility and its antibacterial activity is intricate and can be influenced by a myriad of other factors, including pH, temperature, and the precise bacterial strain under scrutiny.

Consequently, a continuous and evolving research endeavor is imperative to comprehensively fathom and refine this facet of chitosan’s antibacterial attributes. This pursuit is geared toward unraveling the nuanced dynamics and optimizing the interplay between solubility and antibacterial potency, enabling the harnessing of chitosan’s potential across diverse applications.

### 4.5. Effect of pH on Chitosan Solubility and Its Antibacterial Activity

We have already emphasized that even high-molecular-weight chitosan in acidic environments becomes soluble. This solubility exposes its positively-charged amino groups (−NH_3_^+^), which interact with the negatively-charged components of bacterial cell walls. As a result, it damages the bacterial cell wall, disrupting metabolism and leading to the death of the bacteria. Therefore, chitosan’s antibacterial activity is high under acidic conditions [136].

Conversely, when confronted with higher pH levels, the solubility of chitosan diminishes, subsequently curbing its potential to engage with bacterial cells. Consequently, as the environmental pH escalates, the antibacterial efficacy of chitosan experiences a decline.

The permeability of the bacterial cell wall stands as a pivotal determinant in the efficacy of antibacterial agents. This significance stems from the composition and role of the bacterial cell wall, which serves as a protective barricade against detrimental agents. Amplifying the permeability of the bacterial cell wall holds the consequence of rendering it more penetrable to antibacterial agents, enabling them to more effectively infiltrate the bacterial cell and perturb its functions. This mechanistic principle frequently underlies the mode of action for numerous antibiotics and antimicrobial substances [137].

In the case of chitosan, an increase in the permeability of the bacterial cell wall is one of the key mechanisms by which it exerts its antibacterial activity. The positively-charged amino groups of chitosan can interact with the negatively-charged components of the bacterial cell wall, leading to increased permeability [138]. This increased permeability disrupts the balance of substances entering and exiting the bacterial cell, leading to cell damage and death [139,140]. It is important to note that different bacterial species may have varying levels of cell wall permeability, influencing their susceptibility to different antibacterial agents, including chitosan.

Chitosan can affect bacterial metabolism, a crucial aspect of its antibacterial activity. When chitosan interacts with bacterial cells, it can disrupt normal metabolic processes that are essential for the survival and proliferation of the bacteria [140]. The disruption of these processes can occur in several ways. For example, chitosan can interfere with the synthesis of proteins and nucleic acids, impede energy production processes, or disrupt the balance of ions across the cell membrane. Each of these impacts can lead to a decrease in the bacterial cell’s viability [141,142]. Moreover, chitosan’s ability to increase the permeability of the bacterial cell wall can also contribute to these metabolic disruptions. This is because an increase in cell wall permeability can lead to an imbalance in the flow of substances into and out of the cell, which can disrupt normal cellular metabolism [142].

Undoubtedly, the coagulation process assumes a role in chitosan’s antibacterial efficacy. Coagulation, characterized by the aggregation of particles, can be provoked when chitosan interacts with bacterial cells. Under acidic conditions, chitosan has the capacity to trigger the coagulation of proteins. These coagulated protein clusters wield the potential to disrupt the integrity of the bacterial cell membrane, hamper the microorganism’s metabolic functions, or even culminate in cell lysis [143]. Furthermore, the aggregation of coagulated proteins can give rise to a barricade that obstructs bacteria from accessing essential nutrients or expelling waste products. This consequential hindrance to vital processes can significantly contribute to the bacteria’s eventual demise.

## 5. The Most Important Chitosan-Based Antibacterial Materials

### 5.1. Films

Chitosan-based films in their pure form possess innate antibacterial properties and biocompatibility, rendering them adaptable for a range of uses. The inherent antibacterial quality of these films plays a pivotal role in mitigating bacterial growth within food products, consequently extending their shelf life and upholding the standard of food quality. As a consequence, these films are frequently harnessed in the context of food packaging, underlining their significance in bolstering food safety and minimizing waste [89,90,144,145,146,147,148,149].

In the biomedical field, pure chitosan films serve as effective wound dressings due to their biocompatibility, ability to promote wound healing, and antibacterial properties. They also have potential as drug delivery systems, with drugs being incorporated within the film and released over time for sustained therapeutic effect [150,151,152,153].

One limitation of pure chitosan films is their poor mechanical strength and water resistance, but these properties can be improved by blending chitosan with other polymers or by modifying its structure to create chitosan derivatives [154,155].

To address these limitations, derivatives of chitosan can serve as alternatives to pure chitosan. For example, carboxymethyl chitosan (CMC), a water-soluble chitosan derivative, exhibits antibacterial attributes. It is generated through the carboxymethylation process, which augments its solubility and imparts additional advantageous traits [156]. CMC films are made using solution casting and are often used in food packaging due to their antibacterial properties, good mechanical properties, and transparency [157]. These traits help improve food safety and extend shelf-life. Such effective systems are described for a range of perishable vegetables, fruits, meats, and fish and include both regular and edible packaging [158]. Moreover, CMC films are applied in the biomedical field, particularly as wound dressings. They possess excellent moisture retention capabilities and promote wound healing. Moreover, often they are smooth and porous and characterized by good adhesion [159].

Chitosan ester-based and amide films are acknowledged for their heightened hydrophobicity in contrast to chitosan, thereby augmenting their resistance to water. This characteristic expands their utility across domains where resilience to moisture is imperative, such as in food packaging. In this context, these films can play a crucial role in prolonging shelf life and preserving food quality [160,161]. In addition, the mentioned chitosan films retain the inherent antibacterial properties of chitosan, making them effective in preventing bacterial growth [162]. This is especially useful in applications like wound dressings where infection prevention is crucial [163]. Moreover, the films can be used as drug delivery systems, with the drug incorporated within the film and released over time [164]. Similar properties are exhibited by chitosan–poly(vinyl alcohol) [165], chitosan–poly(acrylic acid) [166], chitosan–polyethylene glycol [167], and chitosan–alginate systems [103].

### 5.2. Nanoparticles

Chitosan nanoparticles are the foundational structures that exhibit augmented antimicrobial activity against a range of bacterial and fungal species [168]. Different degrees of deacetylation and molecular weights of chitosan have been shown to interact synergistically with sulfamethoxazole, an antibiotic, enhancing its activity against *P. aeruginosa* [169]. The delivery of antibacterial drugs, such as levofloxacin or clarithromycin, using chitosan nanoparticles has demonstrated significant potential in combating drug-resistant pathogens, like methicillin-resistant *S. aureus* and the multidrug-resistant *A. baumannii* [170,171]. Similarly, chitosan-based nanoparticles were demonstrated to kill ampicillin-resistant bacterial pathogens [172].

Expanding upon this, chitosan nanoparticles can be combined with metals to enhance their antimicrobial potential. Notably, chitosan–silver nanoparticles have demonstrated synergistic effects when used with antibiotics against various bacterial strains [173]. This potent antimicrobial action has been employed for combating bacterial diseases in aquaculture [174]. Similarly, gold–chitosan nanoparticles have shown promise as robust antimicrobial agents against *E. coli* and *S. aureus* [175,176], while zinc–chitosan nanoparticles have exhibited remarkable antibacterial properties against Gram-positive and Gram-negative bacteria [177,178].

Chitosan nanocomposites, which combine chitosan with other materials like polymers or biocompatible materials, offer a synergistic effect. For instance, chitosan nanocomposites with carbon nanotubes [179] have shown potential for wastewater treatment due to their improved antibacterial action against *E. coli*. Silver/laterite/chitosan nanocomposites similarly are characterized as having high in vitro and in vivo antibacterial activity toward *S. aureus* and *P. aeruginosa*. In the medical field, chitosan–collagen nanocomposites have been used in wound healing owing to their antibacterial and biocompatible properties [180].

Chitosan can undergo chemical modifications to yield derivatives with customized attributes. These modifications often manifest in the form of nanoparticles, which have exhibited amplified antimicrobial effectiveness against specific pathogens. For instance, carboxymethyl chitosan nanoparticles have showcased heightened antimicrobial prowess against *S. aureus* and *E. coli*. Additionally, chitosan nanoparticles that have been grafted with polyethylene glycol have presented promising outcomes in enhancing the antimicrobial efficacy of antibiotics like ciprofloxacin.

Lastly, chitosan nanocapsules have emerged as potent delivery vehicles for antibiotics. By encapsulating the drug, these nanocapsules can shield it from premature degradation, thus improving its stability and bioavailability. For instance, chitosan nanocapsules loaded with ampicillin have shown improved antimicrobial action against both Gram-positive and Gram-negative bacteria [181,182,183,184].

The versatility of chitosan-based nanoparticles, coupled with their intrinsic antimicrobial attributes, positions them as a propitious material in the battle against bacterial infections. This stance will propel the advancement of antimicrobial strategies across diverse domains, encompassing medicine, environmental science, and aquaculture.

### 5.3. Chitosan-Based Nonwoven Materials

Chitosan-based nonwoven materials present an appealing format for harnessing chitosan’s innate antimicrobial capacity. These materials are extensively employed in various medical contexts, including wound dressings, face masks, and surgical gowns. In the realm of wound care, for instance, dressings crafted from chitosan-based nonwoven materials serve a dual purpose: they offer a tangible barrier against infections while concurrently engaging in active antibacterial combat. This synergy not only fosters wound healing but also accelerates the recuperation process [185].

The effectiveness of chitosan can be further enhanced by creating composite nonwoven materials, where chitosan is combined with other polymers or biocompatible substances. An example is chitosan–poly(lactic acid) composites, which have been utilized in applications that require superior mechanical strength and stability, like durable wound dressings and healthcare textiles [150,186,187]. Combining chitosan with cellulose has also been shown to improve moisture management properties, creating nonwoven materials that are more comfortable for patients [188].

In addition to chitosan itself, derivatives of chitosan can be employed to manufacture nonwoven materials with improved properties. For instance, carboxymethyl chitosan nonwoven materials exhibit enhanced hydrophilicity, making them suitable for applications where moisture management is vital, such as in wound dressings and incontinence products [189]. Additionally, chitosan–graft–poly(lactic acid) nonwoven materials have demonstrated excellent moisture retention and improved comfort, making them effective for personal care products and advanced wound-care applications [190]. Chitosan-based nonwoven materials can also be fortified with nanoparticles to augment their antibacterial capabilities. For example, the integration of silver nanoparticles into chitosan nonwoven materials has been shown to enhance their antimicrobial properties significantly [191,192]. Similarly, the incorporation of zinc oxide nanoparticles and curcumin can provide additional antibacterial and UV-protective qualities, making these materials suitable for outdoor clothing and sun-protective textiles [193]. Moreover, chitosan can serve as a vehicle for incorporating antibiotics into nonwoven materials, providing sustained and targeted antimicrobial action. For example, nonwoven materials containing tetracycline-loaded chitosan have been used in wound dressings for severe or chronic wounds, resulting in improved wound healing rates [194].

We hold the conviction that the versatility inherent to chitosan and its derivatives renders them an exceptional selection for formulating an extensive array of antibacterial nonwoven materials. As this field continues to progress, the ongoing advancements are poised to yield even more potent solutions in the realm of managing bacterial infections, thereby fostering enhanced health and overall well-being.

### 5.4. Hydrogels

Even simple chitosan-based hydrogels represent effective multifunctional (including antibacterial) materials. Their creation leverages the natural antimicrobial properties of chitosan, and they have found a broad range of applications in the medical field. For example, in drug delivery systems, these hydrogels can effectively release antimicrobial agents over a prolonged period, assisting in combating infections in a controlled manner [195]. Furthermore, in wound-care applications, chitosan hydrogels can provide a moist environment that not only prevents infection but also promotes wound healing [196,197].

Composite chitosan hydrogels incorporate other polymers or biocompatible materials to enhance the properties of chitosan. For instance, chitosan–alginate hydrogels boast high water retention and biocompatibility, making them ideal for applications like contact lenses and tissue-engineering scaffolds [198]. Moreover, combining chitosan with gelatin can result in hydrogels with improved mechanical properties, expanding their usage to areas such as load-bearing tissue repair [199].

Chitosan-derivative-based hydrogels, employing compounds like carboxymethyl chitosan [200] or chitosan–graft–polyethylene glycol [201], bring unique properties to the table. Such hydrogels exhibit improved hydrophilicity and biocompatibility, making them suitable for various biomedical applications. For instance, carboxymethyl chitosan hydrogels, owing to their superior hydrophilicity, can find applications in fields where water management is crucial, such as for contact lenses and moisture-retaining wound dressings [202,203]. Chitosan-based hydrogels can also be synthesized with nanoparticles to further improve their antibacterial capabilities. For example, incorporating silver nanoparticles into chitosan hydrogels can significantly boost their antimicrobial properties [204]. These hydrogels show promise in wound-care applications, providing an effective barrier against a wide range of bacteria. In addition, chitosan gels provide a favorable environment for cell proliferation and the regeneration of damaged tissues [205]. We believe that chitosan-based hydrogels will occupy a large niche in the pharmaceutical market in the near future.

## 6. Conclusions

This comprehensive review has meticulously explored the diverse antibacterial attributes of chitosan, spanning its preparation methods and extensive applications, all of which underscore its remarkable versatility and untapped potential. Nevertheless, the journey toward attaining efficient and sustainable production is not devoid of challenges.

We have meticulously examined chemical, physical, biological, hybrid, and green extraction methods, each presenting distinct merits and limitations. The selection of the appropriate method hinges on a multifaceted interplay involving the desired chitosan properties, resource availability, cost considerations, and environmental impact.

Chitosan’s antibacterial efficacy is intricately linked with its structural and physicochemical characteristics, the specific type of targeted bacteria, and the surrounding environmental conditions. The intricate interrelation between factors such as cationic density, molecular weight, water solubility, pH, and antibacterial potential has been dissected, revealing a complex nexus.

The realm of chitosan-based materials, spanning films, nanoparticles, nonwoven materials, and hydrogels, vividly exemplifies its versatile deployment across domains encompassing food, biomedicine, and agriculture sectors.

Future research endeavors should remain dedicated to refining the sustainability and efficiency of chitosan production, simultaneously propelling the innovation of novel chitosan-based materials and applications.

In summary, the trajectory of chitosan’s future emerges as auspicious, underpinned by the steady march of technological and sustainable advancements. However, orchestrating these strides in harmony with environmental sustainability and efficiency emerges as an indispensable mandate.

## Data Availability

Not applicable.

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
