# Peer review of "Chitosan and Its Derivatives: Preparation and Antibacterial Properties"

_materials, 2023, doi:10.3390/ma16186076_

Round 1
Reviewer 1 Report
The manuscript under consideration is a review of chitosan-based antibacterial systems, as well as related issues (sources of raw materials, production methods). The topic seems important, such reviews are valuable for choosing directions for further research, but there are remarks about this one.
It is somewhat perplexing to put acidic hydrolysis on the same level as alkaline hydrolysis, as methods for chitin deacetylation. Treatment with strong acids is used for crustacean shells, but for dissolving the carbonate component of the exoskeleton rather than for the hydrolysis of amide bonds. It is also disputable to put physical methods on the same level as chemical ones, rather than as auxiliary ones. Chitin-to-chitosan conversion requires a (bio)chemical reaction, and physical processing can only assist it.
The authors’ choice of publications to illustrate certain statements is sometimes debatable. E.g., they write: “The alkali breaks the acetyl-amino linkage in chitin, causing the release of acetate ions and the formation of chitosan [17]” (lines 118 and 119), but Ref. [17] is of 2014, whilst alkaline chitin hydrolysis was discovered much earlier.
Line 202: “UV and Gamma Irradiation” — these are rather different orders of energy, so their combination looks strange. Gamma radiation breaks chemical bonds in a random way, it is suitable for creating free radical sites (for grafting) in a polymer, but it is not suitable for breaking a specific bond.
There are a number of questions about Fig. 3. The side chain in [100] looks like a polyketone. [93] — is it cross-linked chitosan? But how does this compare with the statement of low-molecular-weight chitosan having greater antibacterial activity? In all cases, derivatization occurs at the amino group, which makes it difficult or even eliminates the possibility of its protonation. How does this compare with the polycationic theory of the biological activity of chitosan? Is the amino group “sacrificed” for the sake of attaching something much more active or valuable, relegating chitosan to the role of just a substrate? It would be more logical to carry out derivatization by the hydroxylic group, as shown in Scheme 3, but where at least one example in Fig. 3?
Let’s look at some contradictions. Line 247: “Solvent-free Processes”, and line 250: “These fluids can act as both a solvent and a deacetylating agent”. So is there a solvent or not? Lines 252 and 253: “…can result in chitosan with high purity [37]”, but Ref. [37] refers to chitin extraction. Lines 506 and 507: “…the penetration of external infections into the bacterial cell” — do bacteria themselves suffer from infections?
Repetitions are striking. “Microwave” is mentioned in Subsects 3.3.3 and 3.5.3. Biological methods: Subsects 3.2 and 3.5.4. Line 364: “polycation macromolecule” (a polycation is a special case of a macromolecule). The interaction of a positively-charged polycation with negatively-charged bacterial walls is mentioned several times: on pages 8, 9, 15, and 16 (and only here is a reference to the source [131]).
The manuscript is not free from typos and inaccuracies. E.g., chitosan is called a polysaccharide (lines 24, 43, 114, and partly 450), although it is an aminopolysaccharide, and it is just the amino group which is responsible for its considered (antibacterial) properties. Line 63: probably, “No allergenicity”? (by the way, what causes allergenicity: chitosan itself or its impurities?) Line 171: “speed” — maybe, “rate”? Lines 357 and 358: “…the interaction of a positive macromolecule with the negative surface of the cell” — maybe, “positively-charged” and “negatively-charged” (as in lines 363 and 364)? Lines 393–395: the phrase is not finished. Scheme 2: are the red and blue R radicals different only in the place of attachment, or in chemical nature as well? Scheme 3: styrene is drawn (no oxygen) instead of the declared benzaldehyde. Lines 586–588: “Acyl group length impacting antibacterial efficacy noticeable when curtailing E. coli growth using acyl chitosan derivatives” — no personal verb. Line 684: “In addition, chitosan the mentioned films retain the inherent antibacterial properties…” — a preposition is missing.
Throughout the whole review, the authors attach no importance to the nature of the acid which chitosan is dissolved in (should be dissolved) and pass over in silence the so-called “soft derivatization”, which consists in dissolving chitosan in biologically active acids in order to preserve the protonated amino groups and add the effect(s) of the anions. There are a number of interesting works in this direction, without which the review looks incomplete.
Since the authors used such an exalted word as “aurora” with the adjective “new” in the title, they should have touched on the history of the discovery of chitosan, the discovery of its antimicrobial properties (the priority!) and the bibliometry of publications, as well as to estimate what (approximately, of course) percentage of research developments have already been commercialized. This is just our recommendation to authors, not a remark.
The manuscript can be published after the noted shortcomings are eliminated.

Author Response
First, the authors are deeply and sincerely grateful to Reviewers and Editor for their unselfish and extremely important work, for thoroughly checking the manuscript, for valuable comments and advice, which have significantly improved the starting text of the submitted manuscript. Our responses follow below.
It is somewhat perplexing to put acidic hydrolysis on the same level as alkaline hydrolysis, as methods for chitin deacetylation. Treatment with strong acids is used for crustacean shells, but for dissolving the carbonate component of the exoskeleton rather than for the hydrolysis of amide bonds. It is also disputable to put physical methods on the same level as chemical ones, rather than as auxiliary ones. Chitin-to-chitosan conversion requires a (bio)chemical reaction, and physical processing can only assist it.
- Corrected
The authors’ choice of publications to illustrate certain statements is sometimes debatable. E.g., they write: “The alkali breaks the acetyl-amino linkage in chitin, causing the release of acetate ions and the formation of chitosan [17]” (lines 118 and 119), but Ref. [17] is of 2014, whilst alkaline chitin hydrolysis was discovered much earlier.
- The authors agree with the remark, but their experience is that many reviewers require the use of newer references than direct works when a particular phenomenon has been discovered
Line 202: “UV and Gamma Irradiation” — these are rather different orders of energy, so their combination looks strange. Gamma radiation breaks chemical bonds in a random way, it is suitable for creating free radical sites (for grafting) in a polymer, but it is not suitable for breaking a specific bond.
- Corrected
There are a number of questions about Fig. 3. The side chain in [100] looks like a polyketone. [93] — is it cross-linked chitosan? But how does this compare with the statement of low-molecular-weight chitosan having greater antibacterial activity? In all cases, derivatization occurs at the amino group, which makes it difficult or even eliminates the possibility of its protonation. How does this compare with the polycationic theory of the biological activity of chitosan? Is the amino group “sacrificed” for the sake of attaching something much more active or valuable, relegating chitosan to the role of just a substrate? It would be more logical to carry out derivatization by the hydroxylic group, as shown in Scheme 3, but where at least one example in Fig. 3?
- 100 is not a polyketone, it is chitosan quaternized by the amino group. 93 is not cross-linked chitosan, it is chitosan to which a sulfanilamide-like drug is conjugated. The structural fragment SO2-NH-R refers to the grafted sulfanilamide substituent, and not to the polymer backbone. NH-R is not the second chitosan macromolecule. Figure 3 does not contain random connections and errors
Let’s look at some contradictions. Line 247: “Solvent-free Processes”, and line 250: “These fluids can act as both a solvent and a deacetylating agent”. So is there a solvent or not? Lines 252 and 253: “…can result in chitosan with high purity [37]”, but Ref. [37] refers to chitin extraction. Lines 506 and 507: “…the penetration of external infections into the bacterial cell” — do bacteria themselves suffer from infections?
- Corrected.
Chitosan is a continuation of chitin, and if chitin is of high purity, then chitosan is also
Repetitions are striking. “Microwave” is mentioned in Subsects 3.3.3 and 3.5.3. Biological methods: Subsects 3.2 and 3.5.4. Line 364: “polycation macromolecule” (a polycation is a special case of a macromolecule). The interaction of a positively-charged polycation with negatively-charged bacterial walls is mentioned several times: on pages 8, 9, 15, and 16 (and only here is a reference to the source [131]).
- Corrected. About “Microwave” our consultants did not reveal any sedition. We will be grateful to the reviewer if he allows us to leave the "microwaves" as they are
The manuscript is not free from typos and inaccuracies. E.g., chitosan is called a polysaccharide (lines 24, 43, 114, and partly 450), although it is an aminopolysaccharide, and it is just the amino group which is responsible for its considered (antibacterial) properties. Line 63: probably, “No allergenicity”? (by the way, what causes allergenicity: chitosan itself or its impurities?) Line 171: “speed” — maybe, “rate”? Lines 357 and 358: “…the interaction of a positive macromolecule with the negative surface of the cell” — maybe, “positively-charged” and “negatively-charged” (as in lines 363 and 364)? Lines 393–395: the phrase is not finished. Scheme 2: are the red and blue R radicals different only in the place of attachment, or in chemical nature as well? Scheme 3: styrene is drawn (no oxygen) instead of the declared benzaldehyde. Lines 586–588: “Acyl group length impacting antibacterial efficacy noticeable when curtailing E. coli growth using acyl chitosan derivatives” — no personal verb. Line 684: “In addition, chitosan the mentioned films retain the inherent antibacterial properties…” — a preposition is missing.
- Corrected.
Regarding the allergenicity of chitosan: chitosan itself is not characterized by allergenicity, it can be caused by impurities, in particular residual melanin.
Regarding Scheme 2: the radicals are different in chemical nature: O-substitution in red and N-substitution in blue
Throughout the whole review, the authors attach no importance to the nature of the acid which chitosan is dissolved in (should be dissolved) and pass over in silence the so-called “soft derivatization”, which consists in dissolving chitosan in biologically active acids in order to preserve the protonated amino groups and add the effect(s) of the anions. There are a number of interesting works in this direction, without which the review looks incomplete.
- Yes, the authors agree with the reviewer and admire his insight. This is indeed a very important question. This is evidenced by the entire experience of the authors in the chemistry of chitosan. But this is a very complex and ambiguous question. The authors did not intend to discuss such a question. The issue of acid requires a separate review. Most of the works cited use an acetic acid solution
Since the authors used such an exalted word as “aurora” with the adjective “new” in the title, they should have touched on the history of the discovery of chitosan, the discovery of its antimicrobial properties (the priority!) and the bibliometry of publications, as well as to estimate what (approximately, of course) percentage of research developments have already been commercialized. This is just our recommendation to authors, not a remark
- The title is corrected
Reviewer 2 Report
Dear author,
The review entitled “Chitosan And Its Derivatives: A New Aurora in Antibacterial Systems” has been intensively reviewed and evaluated. Although present study was considered an interesting study, there were some points that need to be revised. Hereby, I would like to present my suggestions and revisions.
Revision_1: (line 6) acronyms should be added after the full name
Revision_2: (line 15; line 36) writing that chitosan is soluble in water is an understatement. I suggest authors specify the pH range
Revision_3: (line 84) Correct the number at the beginning of the line
Revision_4: In the paragraph "3.3. Physical Methods" check the list
Revision_5: In my opinion the authors have gone too far to describe the methods of obtaining chitosan. This is not the focus of the work; so, the authors must fix the whole section or change the title of the paper.
Revision_6: No reference is made to the gelation of chitosan with cyclodextrins in any part of the work. Regarding the increase in antibacterial properties, I suggest this paper (Pharmaceutics 2021, 13, 1293. https://doi.org/10.3390/pharmaceutics13081293) to describe this aspect and also add the complexation of chitosan with cyclodextrins.
Revision_7: I think it is necessary to expand the introduction by adding some examples and some references on the extraordinary activities of chitosan. Because in this way the introduction is insufficient and focuses on topics off topic. I suggest below some examples that authors should consider (Critically important request):
- biosensor activity ”https://doi.org/10.1016/j.matpr.2023.01.123
- treatment of carcinogenic pathologies and specific targeting “Pharmaceutics 2022, 14, 942. https://doi.org/10.3390/pharmaceutics14050942”
- antioxidant “DOI: 10.1016/B978-0-12-800268-1.00002-0”
- antifungal activity “doi: 10.3390/ijms20020332”
- neurodegenerative diseases “Pharmaceuticals 2022, 15, 1206. https://doi.org/10.3390/ph15101206”
After these observations the work can be published
Author Response
First, the authors are deeply and sincerely grateful to Reviewers and Editor for their unselfish and extremely important work, for thoroughly checking the manuscript, for valuable comments and advice, which have significantly improved the starting text of the submitted manuscript. Our responses follow below.
Revision_1: (line 6) acronyms should be added after the full name
- We share the view of the reviewer, however, according to the order of the Rector of RUDN University, the University is the official name of the university and we do not have the right to write it differently (acronyms, decoding, and so on), we must write only "RUDN University"
Revision_2: (line 15; line 36) writing that chitosan is soluble in water is an understatement. I suggest authors specify the pH range
- In these phrases, a problem is posed; this is a designation of a problem, not a statement. A detailed discussion of the pH value at which chitosan dissolves, as well as a number of factors affecting its solubility, the reader will find in paragraph 4 (in particular section 4.5)
Revision_3: (line 84) Correct the number at the beginning of the line
- Corrected
Revision_4: In the paragraph "3.3. Physical Methods" check the list
- Сhecked out
Revision_5: In my opinion the authors have gone too far to describe the methods of obtaining chitosan. This is not the focus of the work; so, the authors must fix the whole section or change the title of the paper.
- Corrected (the title has been specified)
Revision_6: No reference is made to the gelation of chitosan with cyclodextrins in any part of the work. Regarding the increase in antibacterial properties, I suggest this paper (Pharmaceutics 2021, 13, 1293. https://doi.org/10.3390/pharmaceutics13081293) to describe this aspect and also add the complexation of chitosan with cyclodextrins.
- Corrected
Revision_7: I think it is necessary to expand the introduction by adding some examples and some references on the extraordinary activities of chitosan. Because in this way the introduction is insufficient and focuses on topics off topic. I suggest below some examples that authors should consider (Critically important request):
- biosensor activity ”https://doi.org/10.1016/j.matpr.2023.01.123
- treatment of carcinogenic pathologies and specific targeting “Pharmaceutics 2022, 14, 942. https://doi.org/10.3390/pharmaceutics14050942
- antioxidant “DOI: 10.1016/B978-0-12-800268-1.00002-0
- antifungal activity “doi: 10.3390/ijms20020332”
- neurodegenerative diseases “Pharmaceuticals 2022, 15, 1206. https://doi.org/10.3390/ph15101206”
- Corrected
After these observations the work can be published
- Thank you
Reviewer 3 Report
In this review, the authors reported “Chitosan and Its Derivatives: A New Aurora in Antibacterial
Systems”. However, the author should address the following comments before recommending it for publication in “materials”
Some comments
1. In my opinion, the review article does not contain updated or novel information except for some parts.
2. Moreover, this review cited more review articles instead of research articles. What is the need to present well-known information about isolation methods in a very usual style?
3. The source of chitosan was reported well in a previously published review. Here authors present the same information in a running style of writing. But it was not easy to follow.
4. Figure 2. Check the subheading carefully. Please write “Alkaline deacetylation”. What does mean “hybrid methods” and “green extraction”. Please check and present more appropriately.
5. Please check the research articles related to chitosan and its antibacterial applications. For example, https://doi.org/10.1016/j.ijbiomac.2023.124129; https://doi.org/10.1016/j.ijbiomac.2023.125087.
Minor editing of English language required
Author Response
First, the authors are deeply and sincerely grateful to Reviewers and Editor for their unselfish and extremely important work, for thoroughly checking the manuscript, for valuable comments and advice, which have significantly improved the starting text of the submitted manuscript. Our responses follow below.
- In my opinion, the review article does not contain updated or novel information except for some parts.
- In the opinion of the authors and other reviewers, the authors have fulfilled the tasks that they set in their review. There are other reviews in the literature that suit the tastes of other readers. We must give the reader a choice
- Moreover, this review cited more review articles instead of research articles. What is the need to present well-known information about isolation methods in a very usual style?
- The authors believe that the balance between classics and innovation (between highly cited time-tested articles and new publications), on the contrary, is the strong side of the presented review. Other reviewers did not comment on this. Again, we must give the reader a choice.
- The source of chitosan was reported well in a previously published review. Here authors present the same information in a running style of writing. But it was not easy to follow.
- If you look, then everything has analogues, including ANY published review. However, each review is the work of the author, where he focuses the reader's attention on the most important points from his point of view. In addition, the author expresses his attitude to the problem under review. In this regard, each review is unique. The authors of this review believe that it has many characteristic features inherent only to it and has its own style and will definitely find its readership for whom it will be important, interesting and even inspiring
- Figure 2. Check the subheading carefully. Please write “Alkaline deacetylation”. What does mean “hybrid methods” and “green extraction”. Please check and present more appropriately.
- The authors are very grateful to the reviewer for their attentiveness and meticulousness! Thank you very much! We corrected
- Please check the research articles related to chitosan and its antibacterial applications. For example,https://doi.org/10.1016/j.ijbiomac.2023.124129; https://doi.org/10.1016/j.ijbiomac.2023.125087
- The authors thank the referee for interesting and inspiring publications. We have cited them in the revised version of the review
Minor editing of English language required
- Corrected
Author Response
First, the authors are deeply and sincerely grateful to Reviewers and Editor for their unselfish and extremely important work, for thoroughly checking the manuscript, for valuable comments and advice, which have significantly improved the starting text of the submitted manuscript. Our responses follow below.
- English needs improvement through out the manuscript. There are many spelling mistakes and some phrases are not clear in the original paper.
- Corrected
- In the Introduction, the novelty is weakly emphasized.
- Corrected
- Why do only three figures appear in this manuscript, considering that it is a review? What is the explanation?
- Unfortunately, the Journal does not require a minimum number of figures. The authors do not want to argue about this issue. I will only say that I personally know readers who are annoyed by the abundance of figures, especially in reviews. Such readers have told me that they want to draw information from the text, and not look at the pictures. We must give readers a choice. The authors consider the number of figures sufficient and will be very deeply grateful to the reviewer if he allows not to introduce additional figures into the text.
- A timeline is suggested to be added to show the development of the field
- For this review, we see no point in this. We will be grateful to the reviewer if he does not consider this requirement mandatory
- Abbreviations are missing in this manuscript
- Again, we have the right to write without abbreviations
- There are some interesting conclusions drawn but insufficient focus on future work
- We think that is enough. This is not a philosophical treatise.
We are very grateful to Referee for his remarks, but at the same time we very much hope that he allows us to have our own view and our own attitude to our work. We have corrected all the critical remarks.
Round 2
Reviewer 1 Report
First of all, it should be noted that the authors have changed the numbering of references in Fig. 3, without reflecting this fact in yellow. I was asking questions using the old numbering, while the authors answered me using their new numbering. Well, let's use the new one. Ref. 99 shows chitosan hydrogel (crosslinked with glutaraldehyde). But it is the hydrogel with zinc phthalocyanine-colistin (ZnPc-Col) conjugate which has an antibacterial effect against Pseudomonas aeruginosa, rather than just the pure hydrogel (with a significantly reduced content of amino groups). Ref. 106: An oxygen atom in the backbone inside the parentheses is clearly missing, making the polyester look like a polyketone. So, the authors' statement "Figure 3 does not contain random connections and errors" seems unfounded. All the structures on Fig. 3 need close checking.
Author Response
Sorry for the misunderstanding! Figure 3 corrected.
Reviewer 2 Report
The requested changes have been made, the work can be published.
Author Response
Thank you!
Reviewer 3 Report
I could not find the revised review articles instead of the two-page reviewer comments from the "Download manuscript".
-
Author Response
We are very sorry that you could not find the file! Apparently there was a technical error, we are uploading an updated manuscript. We hope this time the file will be in place!